# Reproducibility Study of "FairViT: Fair Vision Transformer via Adaptive Masking"

## Abstract

Vision Transformers (ViTs) have recently demonstrated strong performance in computer vision tasks but often encounter challenges related to fairness for attributes like gender and hair colour. FairViT by Tian et al. (2024), aims to address this challenge, by introducing adaptive masking combined with a distance-based loss, to improve fairness and accuracy, while maintaining competitive computational efficiency compared to other baseline methods. In our reproducibility study, we evaluated FairViT on the CelebA dataset on tasks related to attractiveness and facial expression prediction, while considering specific sensitive attributes. We then compared FairViT against the Vanilla and Fair Supervised Contrastive Loss (FSCL) baseline models. Contrary to the original claim regarding the effectiveness of adaptive masking, we found its impact to be negligible in terms of both fairness and accuracy, a finding confirmed also on the UTKFace dataset. On the other hand, the distance-based loss demonstrated partial effectiveness, but primarily when evaluated in the context of different architectures. Finally, in terms of computational efficiency, FairViT required approximately twice the training time per epoch relative to the Vanilla model and failed to outperform FSCL, which exhibited the lowest training time for the specified dataset size used by the authors. Overall, our findings highlight the potential effectiveness of the proposed distance loss. However, the adaptive masking method did not deliver the expected improvements while also increasing the computational cost. Our implementation is available at [1].

## 1 Introduction

Vision Transformers (ViT) have emerged as a valuable asset for downstream Computer Vision (CV) applications (e.g., classification and detection), achieving superior performance when pre-trained on large-scale datasets, compared to state-of-the-art (SOTA) Convolutional Neural Networks (CNNs) (Dosovitskiy et al., 2021). However, ViTs pose fairness-related challenges stemming from algorithmic and data biases (Sudhakar et al., 2023), which often originate from datasets containing inaccurate or imbalanced demographic information. These biases can be propagated through the model architecture. For instance, the CelebA dataset (Liu et al., 2015) introduces data bias into the model due to widespread inconsistencies and inaccuracies in its attribute labelling (Lingenfelter et al., 2022). When used in ViT models, this bias can be amplified via the attention mechanism (Mandal et al., 2023), disproportionally emphasising certain features. As a result, predictions can be skewed, disadvantaging underrepresented groups, indicating the need for fairer CV architectures.

Existing approaches for mitigating bias in AI systems include pre-processing, in-processing and post-processing techniques, each applied at different stages of the pipeline (Kamiran & Calders, 2011; Cruz & Hardt, 2024; Hardt et al., 2016). Pre-processing techniques, such as the use of counterfactual data augmentation (Brinkmann et al., 2023), refine the dataset. In-processing techniques manipulate the model's architecture for de-biasing. Such examples include the use of bi-level optimisation for adjusting the data sampling ratios (Roh et al., 2023), directly removing bias from the query matrix (Sudhakar et al., 2023) of a Vision Transformer, or using De-biased Self-Attention to adjust the ViT through adversarial training (Qiang

---

[1] https://anonymous.4open.science/r/FairViT-reproducibility-study-54B0/

et al., 2024). Lastly, post-processing techniques try to make a model fairer by adjusting the classification thresholds after it is trained (Pleiss et al., 2017). Nevertheless, all these methods are susceptible to the trade-off between accuracy and fairness (Wang et al., 2021), highlighting the need for innovative approaches that balance these two properties, while maintaining computational efficiency.

One recent approach to addressing fairness concerns in ViT models, employed by Tian et al. (2024), proposes the FairViT model, which uses adaptive masking combined with a distance-based loss function to increase the accuracy of the Vision Transformer, while preserving competitive fairness performance. Inspired by this research, we aim to reproduce the original experiments and provide several meaningful extensions to examine the robustness of the FairViT model.

Specifically, this work includes the following components:

[**Reproducibility Study**] **Reproduction of the original paper's experiments.** We attempted to reproduce the main experiments presented in the original paper, in order to verify the authors' claims as presented in Section 2. Our findings indicate that only one of the original claims could be partially reproduced.

[**Extended Work**] **Additional datasets.** As the original paper presents results only on the CelebA dataset, we further evaluated the proposed model on the UTKFace dataset (Zhang & Qi, 2017), examining the robustness of the described method in a different setting.

[**Extended Work**] **Ablation study.** We assessed the impact of the adaptive masking independently, without the combined usage of the proposed distance loss, as this ablation was not included in the original work.

[**Extended Work**] **Evaluation of the proposed loss function on additional architectures.** We investigated the extensibility of the proposed distance loss across different model architectures, with the objective of assessing potential accuracy improvements. Specifically, we experimented with a pre-trained ResNet50 (He et al., 2015) and a Graph Neural Network (GNN) (Zhou et al., 2021) for classification tasks.

[**Extended Work**] **Codebase extension and enhancements.** We integrated additional input arguments into the code to facilitate easier ablation experiments. Additionally, we developed data processing scripts and implemented a method for visualising the attention heatmaps using Gradient Rollout (Abnar & Zuidema (2020)).

## 2 Scope of reproducibility

This work investigates the reproducibility of "FairViT: FairVision Transformer via Adaptive Masking" proposed by Tian et al. (2024). The original study introduces a novel approach aimed at balancing fairness and accuracy for Vision Transformers, reporting improved accuracy while maintaining fairness levels comparable to baseline models.

To evaluate these contributions, we focused on verifying the following primary claims made by the authors:

- **Higher Accuracy and Competitive Fairness via Adaptive Masking.** FairViT incorporates adaptive masks with learnable weights into its attention layers. These group-specific masks are dynamically updated during training, addressing biases inherent in the dataset and improving accuracy for underrepresented groups. As a result, FairViT outperforms SOTA ViT models including Vanilla ViT , TADeT-MMD and TADeT (Sudhakar et al., 2023), FSCL+ and FSCL (Park et al., 2022) in terms of accuracy. Additionally, it achieves comparable or even superior fairness metrics, as demonstrated by evaluations of the authors on the CelebA dataset.

- **Further accuracy improvement via a novel distance loss function.** The authors introduce a distance-based loss function designed to improve accuracy when used in conjunction with the standard cross-entropy loss. This loss acts as a regulariser, shifting misclassified points towards the

desired side of the decision boundary, while maintaining correctly classified points on their original side.

- **Comparable computational efficiency.** The authors claim that FairViT achieves a computational cost comparable to that of the baseline models, thereby balancing performance, fairness and efficiency.

## 3 Methodology

To examine the contributions of FairViT, we attempted to reproduce the experimental results reported by Tian et al. (2024). We focused on comparing FairViT with two of the baselines established in the original paper, the Vanilla ViT and FSCL, reporting the models' performance in terms of accuracy, fairness and computational efficiency. In this section, we present a comprehensive overview of the employed methodology, along with the detailed steps followed during the reproducibility process.

### 3.1 Model description

#### 3.1.1 The Fair Vision Transformer

FairViT employs a Data-efficient Image Transformer (DeiT) (Touvron et al., 2021), which shares similarities with the original Vanilla architecture (Dosovitskiy et al., 2021), primarily in its training strategy. The proposed model combines adaptive masking with a novel distance loss in order to improve accuracy while maintaining competitive fairness performance.

The objective of FairViT is to train a model $f$ using samples $(\mathbf{x}, s, y)$, where $\mathbf{x}$ is the input, $s$ is a sensitive attribute, and $y$ is the target label. Let $\theta$ denote the learnable parameters of the model. Formally, the loss function is minimised:

$$\min_{\theta} L\big(f(\mathbf{x}; \theta), s, y\big) \tag{1}$$

During testing, the sensitive attribute s is treated as unknown and the adapative masking logic is not applied in the linear layers.

**Adaptive Masking.** To handle a binary sensitive attribute $s$, the training dataset is partitioned into $G$ distinct subsets. Two groups are defined, corresponding to the presence or absence of the sensitive attribute, and each group is allocated $\lfloor G/2 \rfloor$ parts, ensuring that each group receives an equal number of subsets, given that the sensitive attributes are binary or are treated as such. The training samples are distributed equally between all parts of each sensitive group, while parts belonging to two different sensitive groups may have a different number of samples. For example, if gender is used as the sensitive attribute, each subset assigned to the male group should contain an equal number of samples. However, this number may differ from that of the subsets assigned to the female group. Subsequently, each of the $G$ parts of the input space is attributed to a learnable mask $M_{l,h,i}$, where $l$ denotes the layer of the transformer, $h$ denotes the head of the multi-attention mechanism and $i$ denotes the specific part of the input space.

In all cases presented below, $\odot$ indicates element-wise multiplication. $\tilde{M}_{l,h}$ denotes the mask matrix, which is derived as the summation of all groups' individual masks, weighted by a set of group-specific learnable coefficients $\varsigma_i$, where $i$ denotes the group index:

$$\tilde{M}_{l,h} = \sum_{i=1}^{G} \varsigma_i \, M_{l,h,i} \tag{2}$$

The described distribution of training data and masks in distinct groups with multiple parts, ensures that each $M_{l,h,i}$ receives sufficient and balanced training data and is able to adapt on the specific characteristics of its allocated part during training. Thus, this setup enables the masks to capture group-specific characteristics and facilitates the learning of diverse features within each sensitive attribute group, as multiple masks and parts are allocated to each attribute.

Upon closer inspection of the code implementation, we observed that masking is involved in multiple components of the architecture. We present the specific way it is implemented below, in order to provide a more informative view of the overall flow of the proposed method:

The masking method is applied to the weights of all linear layers included in the transformer block. Thus, it appears in four different parts of the transformer block, both in its attention module and its Multi-Layer Perceptron (MLP) module. Within the attention module the masking is applied firstly in the linear projection of the input $\mathbf{x}$ which yields the queries, keys and values matrices. It is then involved again in the linear projection of the output of the attention operation ($\mathbf{Attn_{l,h}} = softmax(\frac{\mathbf{QK}^T}{\sqrt{d}})\mathbf{V}$). Finally, masking is applied twice within the transformer blocks' MLP module, as part of its two fully connected layers.

In order to obtain the queries, keys and values, as in the common transformer architecture, a linear layer is used, mapping an input $\mathbf{x} \in R^D$ to an output $\in R^{3D}$, which contains the query $\mathbf{q}$, key $\mathbf{k}$ and value $\mathbf{v}$ vectors, each of dimensionality $D$. Here, the mask matrix $\mathbf{M}$ belongs to the space $R^H$, where $H$ denotes the number of the distinct attention heads in multi-head attention. To obtain each of the vectors $\mathbf{q}$, $\mathbf{k}$ and $\mathbf{v}$, the masking value corresponding to each head is repeated $D/H$ times and multiplied element-wise with the weights of the linear layer. It can be noted that this is equivalent to multiplying each group of $D/H$ output features of a simple linear layer with their corresponding mask value (the mask responsible for these features).

In the case of the linear projection layer, applied to the self-attention outputs, the outputs $\mathbf{Attn_{l,h}}$ are re-weighted by the adaptive masking mechanism for each transformer layer $l$ and head $h$. More specifically, the linear projection layer maps its input $\mathbf{x} \in R^D$ to an output of the same dimensionality, and the adaptive mask is a matrix $\mathbf{M} \in R^D$ multiplied element-wise with the linear layer's weight matrix of dimensionality $D \times D$ (after proper broadcasting). This is equivalent with multiplying each output feature of the linear layer with the corresponding element in the original mask. This leads to the resulting head attention (HA) which is also described in the original paper, and is defined as follows:

$$\text{HA}(x, M_{l,h}) = Linear(\tilde{M}_{l,h} \odot \text{Attn}_{l,h}(x)), \tag{3}$$

which matches the formulation the authors provide.

Subsequently, the single-head results are concatenated as usually done in the context of multi-head attention. Finally, the fully connected layers of the MLP module map an input $\mathbf{x} \in R^{D_1}$ to $R^{D_2}$. The masking operation in this case is a matrix $\mathbf{M} \in R^{D_2}$ that is multiplied element-wise with the output features of each layer. Equivalently, with proper broadcasting, the mask tensor is multiplied element-wise with the $D_1 \times D_2$ weights of each fully connected layer.

**Distance Loss.** In addition to the standard cross-entropy loss used for classification tasks, FairViT uses a distance loss, which encourages misclassified points to move closer to the correct side of the decision boundary, and ensures that correctly classified instances are preserved. A logistic regression model is fitted on the validation set, using the following features: $\hat{y}$, which is the logit of the correct label, and $\hat{y}_k$, which represents the cumulative sum of the logits of the top-$k$ incorrect labels. The goal is to predict the probability of correct classification of each sample by the ViT.

The decision boundary defined by this model is the following:

$$\hat{y} + \omega\,\hat{y}_k + \beta = 0, \tag{4}$$

Afterwards, the sigmoid function, $S(u) = \frac{1}{1+e^{-u}}$, is used to map logits into probabilities $z$. The distance loss $\mathcal{L}_{\text{dist}}$ is thus defined as:

$$\mathcal{L}_{\text{dist}} = \begin{cases} -\gamma\,\Phi(\hat{y}, \hat{y}_k), & \text{if } \hat{y} + \omega\,\hat{y}_k + \beta \geq 0, \\ \Phi(\hat{y}, \hat{y}_k), & \text{otherwise.} \end{cases} \tag{5}$$

where $\gamma$ is a hyperparameter that regulates the influence of the distance $\Phi$ of a point $(\hat{y}, \hat{y}_k)$ to the decision boundary:

$$\Phi(\hat{y}, \hat{y}_k) = \frac{\left|\hat{y} + \omega\,\hat{y}_k + \beta\right|}{\sqrt{1+\omega^2}}. \tag{6}$$

By penalising samples that lie farther from the decision boundary, the loss function encourages the ViT to adjust its weights in order to produce logits that are more likely to yield correct classification results. Combining the cross-entropy loss, $\mathcal{L}_{\text{ce}}$, with the distance loss $\mathcal{L}_{\text{dist}}$, the resulting training loss is the following:

$$\mathcal{L} = \mathcal{L}_{\text{ce}} + \alpha\,\mathcal{L}_{\text{dist}} \quad , \tag{7}$$

where $\alpha$ is a hyperparameter.

### 3.2 Datasets

- **CelebA Dataset:** Owing to its extensive diversity and detailed annotations, the CelebA dataset (Liu et al., 2015) is widely used in fairness research within computer vision. The dataset consists of 202,599 images of 10,177 unique individuals, each of which is annotated for 40 binary attributes including gender, age-related traits and various physical features. These attributes allow for the rigorous evaluation of model biases across diverse demographic groups and are represented as binary values of $\pm 1$. The dataset is publicly available from its official website[2].

- **UTKFace Dataset:** The UTKFace dataset (Zhang & Qi, 2017) was used as a further extension of the original experiments to assess the robustness of the proposed method. It provides a diverse set of over 20,000 facial images, annotated with the attributes of age, gender and ethnicity, making it suitable for evaluating the generalisability of the FairViT approach. The official dataset is available for download at [3].

- **AIDS Dataset:** The AIDS dataset consists of 1,999 graphs and was introduced by Riesen & Bunke (2008). It contains compounds checked for evidence of anti-HIV activity and is thus ideal for binary classification tasks.

- **PROTEINS Dataset:** The PROTEINS dataset consists of 1,113 graphs introduced in Borgwardt et al. (2005). It is used for molecular property prediction and particularly for predicting whether molecules function as enzymes.

### 3.3 Hyperparameters

To reproduce the results of the original paper, we adopted the default training hyperparameters as reported by the authors of FairViT. In the case of FSCL, we used the default hyperparameters set in its official GitHub repository, since no specific hyperparameter settings were provided for this baseline in the FairViT paper. Regarding the ablation study of the distance loss across different architectures, for both the ResNet and the GNN we used the AdamW optimiser. For the former, we opted for a learning rate of 0.0005, while for the latter, a learning rate of 0.01 and a weight decay equal to 0.0005 were used. The complete list of hyperparameters is provided in our GitHub repository.

### 3.4 Experimental setup and code

For reproducing the experiments, we relied on the open-source implementation that the authors provide on GitHub [4]. We modified the codebase in order to be able to also use a Vanilla ViT model initialised with the same pre-trained DeiT weights used by FairViT. For running the FSCL baseline we employed the publicly available source code [5]. We integrated the provided model implementation into our codebase, making only minimal modifications to ensure compatibility with the fairness metrics used in FairViT. It was initially unclear from the original paper whether the ResNet backbone used in FSCL was pre-trained on the full CelebA training set or only on the images of the first 80 individuals. Following correspondence with the authors, we confirmed that only the first 80 individuals were used. However, because FSCL is trained from scratch and does not rely on pre-trained weights - unlike ViT-based models - we report its performance when

---

[2]https://mmlab.ie.cuhk.edu.hk/projects/CelebA.html
[3]https://susanqq.github.io/UTKFace/
[4]https://github.com/abdd68/Fair-Vision-Transformer
[5]https://github.com/sungho-CoolG/FSCL

trained on the full dataset to ensure a fair comparison. We then fine-tuned the classification head of FSCL, using only the images of the first 80 individuals in CelebA, to align our setup with FairViT's experimental design. Additionally, we trained FSCL using the smaller dataset to evaluate its computational cost also in this case. Particularly for the FSCL baseline, we limited our evaluation to a single experiment, as FairViT did not consistently outperform the Vanilla ViT model, so further comparisons with even stronger baselines would be redundant given the computational and environmental cost of experiments.

For experimenting with the CelebA dataset, the authors' implementation dynamically creates a 90-10% train-validation split rather than using the official partitioning. We thus merged the train and validation sets of CelebA and filtered the merged dataset to only keep the images of the first 80 celebrities for training and validation (resulting in approximately 1,700 images), following the original paper. For testing we used the officially provided test set as a whole (19,962 images). We created custom scripts for performing the described data merging and filtering, as these were not included in the provided code.

Since the UTKFace dataset does not include official train, validation and test splits, we performed a custom split into 10,000, 2,400, and 2,400 images for the training, validation, and test set respectively. To assess fairness, we introduced an artificial bias by setting the male-to-female sample ratio to 1:4 for the white race group (1,000 female and 4,000 male samples) and 4:1 for the non-white group (4,000 female and 1,000 male samples). The validation and test sets remained fully balanced.

Regarding the AIDS and PROTEINS datasets, we adopted a standard 80:10:10 split for training, validation and testing respectively. Both datasets were downloaded from the TUDataset repository (Morris et al., 2020).

Additionally, we extended the codebase to include an implementation for calculating the Gradient Attention Rollout (Abnar & Zuidema, 2020) for chosen test images, as it was missing from the original paper's code. For this purpose, we adapted the open source implementation available at [6].

The exact method the authors used for selecting the model for reporting results was not clearly specified. However, in the provided code, we observed that, for each metric, the best score across all epochs is saved and reported in the logs, independently from all other metrics. As we are not certain whether these are the metrics indeed reported in the paper and we argue that this method would not provide a realistic insight into the models' performance, since it does not select one specific trained instance as common for all metrics, we selected the best model for each run based on the highest validation accuracy achieved across all epochs during training.

### 3.5 Evaluation metrics

**Evaluation Metrics.** To evaluate the performance and fairness of the proposed approach, we adopted the same metrics used in the original FairViT study. Specifically, performance was assessed using accuracy, while fairness was evaluated using the following metrics:

- **Balanced Accuracy (BA)**: Adjusts for class imbalance by averaging the recall for each class, measuring how balanced performance is, in terms of accuracy across groups:

$$\text{BA} = \frac{1}{4} \left( \text{TPR}_{s=0} + \text{TNR}_{s=0} + \text{TPR}_{s=1} + \text{TNR}_{s=1} \right) \tag{8}$$

- **Demographic Parity (DP)**: Measures the extent to which the probability of a positive prediction is the same across all sensitive groups:

$$P(\hat{Y} = 1 \mid A = a) = P(\hat{Y} = 1 \mid A = b), \quad \forall a, b \in \text{Sensitive Groups} \tag{9}$$

- **Equalised Opportunity (EO)** (Hardt et al., 2016): Measures how "close" the true positive rate (TPR) is across all sensitive groups:

$$P(\hat{Y} = 1 \mid Y = 1, A = a) = P(\hat{Y} = 1 \mid Y = 1, A = b), \quad \forall a, b \in \text{Sensitive Groups} \tag{10}$$

---

[6]https://github.com/jacobgil/vit-explain.git

### 3.6 Computational requirements

All experiments were conducted using an NVIDIA A100 GPU. Training FairViT for a specific target attribute on the CelebA dataset (30 epochs) required approximately 25 minutes, compared to 20 minutes for the Vanilla ViT model. In total, the reproducibility study - including all ablations and extended experiments - accumulated approximately 40 GPU hours. For Carbon Intensity estimation, we used the 2023 value reported by the European Environment Agency, which states that generating 1 kWh of electricity in the Netherlands results in 263 g $CO_2$-equivalent emissions [7]. All experiments were executed using the Snellius infrastructure located in the Amsterdam Data Tower. The Snellius datacenter reports a Power Usage Effectiveness (PUE) of 1.19 [8]. Using the Machine Learning Emissions Calculator (Lacoste et al., 2019) we estimate that our experiments resulted in approximately 3.2 kg of $CO_2$-equivalent emissions.

## 4 Results

As outlined in Section 2, the original paper presents three main claims. Our study partially validated the second claim regarding the effectiveness of the proposed distance-based loss function. However, we were unable to reproduce neither the first and most important claim nor the third claim related to FairViT's computational efficiency.

### 4.1 Results reproducing original paper

#### 4.1.1 Higher accuracy and competitive fairness via adaptive masking

To verify the first claim, we initialised the FairViT model with pre-trained DeiT weights and fine-tuned it on the CelebA dataset using the methodology proposed in the original study. We then compared its performance with the Vanilla model, a Vision Transformer instance initialised with the same pre-trained DeiT weights and fine-tuned using only the standard cross-entropy loss.

Both models were fine-tuned using images of the first 80 celebrities, following the setup specified by the authors. The three evaluated tasks included predicting expression and attractiveness with gender as the sensitive attribute, and predicting attractiveness with hair colour as the sensitive attribute. The results presented in Table 1 correspond to evaluations conducted on the full test set, replicating the original experimental setup. Moreover, Table 3 presents a replication of the authors' ablation study, offering insights into the effects of using the proposed adaptive masking mechanism in combination with the distance loss.

Contrary to the findings of the original paper, our results did not reveal significant improvements in either accuracy or fairness metrics when the adaptive masks were employed. Specifically, Table 1 demonstrates comparable performance between the Vanilla and FairViT models, with each model performing better in certain metrics but no significant overall improvements observed for FairViT. Furthermore, upon inspecting the results in Table 3, no clear benefits can be attributed to adding a constant mask or subsequently making it trainable.

To further compare the Vanilla and FairViT models qualitatively, we generated Gradient Attention Rollout visualisations for randomly selected test images for the two models, which are available in Figure 1. In contrast to the original study, and although minor differences in the Gradient Attention Rollout may be seen for individual images, we do not observe any substantial differences between the two models, in terms of the relevancy of the regions information is extracted from.

#### 4.1.2 Further accuracy improvement via a novel distance loss function

To validate the second claim, we replicated the ablation study of the original paper and present its results in Table 3. This study focuses, among others, on isolating the impact of the distance-based loss from that of the adaptive masking, allowing independent assessment of its contribution. When comparing the results of $L_{ce}$ with $L_{ce} + L_{dist}$ in terms of accuracy, only slight improvements are observed in two scenarios: (1) expression

---

[7]https://www.eea.europa.eu/en/analysis/indicators/greenhouse-gas-emission-intensity-of-1?activeAccordion=
[8]https://www.clouvider.com/amsterdam-data-tower-datacentre/

Figure 1: Gradient Attention Rollout comparison for Vanilla and FairViT models. Across all three experiements no consistent differences are to be observe between the two models' Gradient Attention Rollouts.

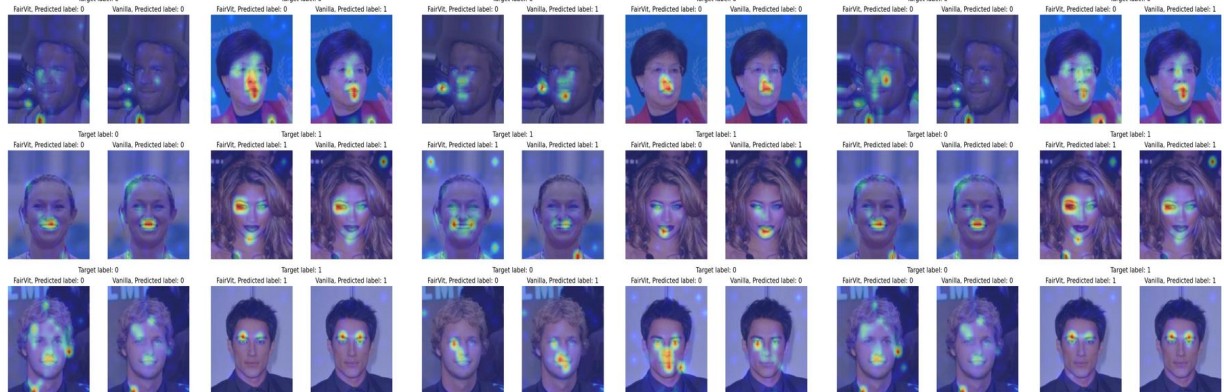

**Y:** Attraction, **S:** Gender          **Y:** Expression, **S:** Gender          **Y:** Attraction, **S:** Hair

Table 1: Accuracy (ACC), Balanced accuracy (BA), Equal Opportunity (EO) and Demographic Parity (DP) for the three classification tasks on CelebA, averaged over three independent runs, for the Vanilla ViT and FairViT models. **Y** denotes the target attribute and **S** denotes the sensitive attribute.

| Model | **Y:** Attraction, **S:** Gender | | | | **Y:** Expression, **S:** Gender | | | | **Y:** Attraction, **S:** Hair Colour | | | |
| | ACC% ↑ | BA% ↑ | $EO_{e-2}$ ↓ | $DP_{e-1}$ ↓ | ACC% ↑ | BA% ↑ | $EO_{e-2}$ ↓ | $DP_{e-1}$ ↓ | ACC% ↑ | BA% ↑ | $EO_{e-2}$ ↓ | $DP_{e-1}$ ↓ |
|---|---|---|---|---|---|---|---|---|---|---|---|---|
| Vanilla | 78.05 ± 0.23 | **73.22 ± 0.28** | **24.80 ± 0.65** | 4.519 ± 0.037 | 89.61 ± 0.34 | 89.19 ± 0.32 | **4.88 ± 0.27** | **1.614 ± 0.040** | **77.94 ± 0.12** | 75.22 ± 0.25 | **4.03 ± 0.56** | 1.822 ± 0.118 |
| FairViT | **78.20 ± 0.25** | 72.82 ± 0.31 | 29.01 ± 0.98 | **4.471 ± 0.117** | **89.71 ± 0.47** | **89.26 ± 0.39** | 5.22 ± 1.03 | 1.655 ± 0.104 | 77.89 ± 0.57 | **75.75 ± 0.52** | 5.43 ± 1.84 | **1.805 ± 0.144** |

Table 2: Accuracy (ACC), Balanced Accuracy (BA), Equal Opportunity (EO), and Demographic Parity (DP) for the classification task **Y:** Attraction, **S:** Gender on CelebA, averaged over three independent runs, for the Vanilla and FairViT models. One run was conducted for the FSCL model.

| Model | **Y:** Attraction, **S:** Gender | | | |
| | ACC% ↑ | BA% ↑ | $EO_{e-2}$ ↓ | $DP_{e-1}$ ↓ |
|---|---|---|---|---|
| Vanilla | 78.05 | 73.22 | **24.80** | 4.519 |
| FSCL | **80.35** | **73.33** | 40.12 | 4.904 |
| FairViT | 78.20 | 72.82 | 29.01 | **4.471** |

prediction with gender as the sensitive attribute (0.21 percentage point increase) and (2) attractiveness prediction with hair colour as the sensitive attribute (0.08 percentage point increase). However, these improvements are marginal and fall within the range of expected statistical variation, making it difficult to draw definitive conclusions regarding the effectiveness of the distance-based loss function. Consequently, we decided to examine this module on a different architecture, as presented in Section 4.2.

### 4.1.3 Comparable computational efficiency

To investigate the third claim, we monitored the time cost per epoch for both the FairViT model and its baselines. The results, presented in Table 4, indicate that FairViT requires nearly twice the training time per epoch compared to the Vanilla model. Therefore, its computational cost is not comparable to that of the Vanilla model, contradicting the authors' claim. Moreover, due to resource constraints, we evaluated FSCL only on the attractiveness prediction task with gender as the sensitive attribute. Using the same dataset as for FairViT (the first 80 celebrities from CelebA) for the contrastive pre-training, resulted in a lower per-epoch training time than FairViT, as shown in Table 5. A substantial increase in per-epoch training time

Table 3: Accuracy (ACC), Balanced accuracy (BA), Equal Opportunity (EO) and Demographic Parity (DP), averaged over three independent runs, for different ablations of the FairViT architecture. **Y** and **S** denote the target and sensitive attribute, $L_{ce}$ and $L_{dist}$ denote the cross-entropy and distance loss and $\Theta$, $\Delta\Theta$ denote the presence of adaptive masks, non-trainable or trainable respectively.

| Model | Y: Attraction, S: Gender | | | | Y: Expression, S: Gender | | | | Y: Attraction, S: Hair Colour | | | |
|---|---|---|---|---|---|---|---|---|---|---|---|---|
| | ACC% ↑ | BA% ↑ | EO$_{\varepsilon-2}$ ↓ | DP$_{\varepsilon-1}$ ↓ | ACC% ↑ | BA% ↑ | EO$_{\varepsilon-2}$ ↓ | DP$_{\varepsilon-1}$ ↓ | ACC% ↑ | BA% ↑ | EO$_{\varepsilon-2}$ ↓ | DP$_{\varepsilon-1}$ ↓ |
| $L_{ce}$ | 78.05 ± 0.23 | 73.22 ± 0.28 | 24.80 ± 0.65 | 4.519 ± 0.037 | 89.61 ± 0.34 | 89.19 ± 0.32 | 4.88 ± 0.27 | 1.614 ± 0.040 | 77.94 ± 0.12 | 75.22 ± 0.25 | **4.03 ± 0.56** | 1.822 ± 0.118 |
| $L_{ce} + \Delta\Theta$ | 77.92 ± 0.21 | 72.53 ± 0.47 | 28.68 ± 2.54 | 4.543 ± 0.046 | 89.86 ± 0.63 | 89.27 ± 0.56 | 6.91 ± 0.57 | 1.812 ± 0.053 | 77.62 ± 0.46 | 75.44 ± 0.83 | 5.85 ± 1.32 | 1.846 ± 0.145 |
| $L_{ce} + L_{dist}$ | 78.02 ± 0.29 | 73.38 ± 0.31 | **23.31 ± 1.72** | 4.528 ± 0.081 | 89.82 ± 0.78 | 89.42 ± 0.68 | **4.45 ± 1.38** | **1.563 ± 0.088** | 78.02 ± 0.29 | 75.22 ± 0.33 | 4.71 ± 1.50 | 1.901 ± 0.188 |
| $L_{ce} + L_{dist} + \Theta$ | **78.24 ± 0.27** | **73.39 ± 0.44** | 25.00 ± 2.75 | 4.519 ± 0.015 | **90.19 ± 0.30** | **89.71 ± 0.20** | 5.39 ± 1.26 | 1.64 ± 0.110 | **78.24 ± 0.27** | **75.81 ± 0.53** | 4.91 ± 1.23 | 1.842 ± 0.100 |
| $L_{ce} + L_{dist} + \Delta\Theta$ | 78.20 ± 0.25 | 72.82 ± 0.31 | 29.01 ± 0.98 | **4.471 ± 0.117** | 89.71 ± 0.47 | 89.26 ± 0.39 | 5.22 ± 1.03 | 1.655 ± 0.104 | 77.89 ± 0.57 | 75.75 ± 0.52 | 5.43 ± 1.84 | **1.805 ± 0.144** |

was observed when using the full CelebA training set for contrastive pre-training. However, this increase is attributable to the larger dataset size rather than architectural differences between the models.

Table 4: Training time per epoch (in seconds), averaged over three independent runs, for Vanilla and FairViT across three classification tasks.

| Model | Y: Attraction, S: Gender | Y: Expression, S: Gender | Y: Attraction, S: Hair Colour |
|---|---|---|---|
| | time (sec) | time (sec) | time (sec) |
| Vanilla | **12.20** | **12.18** | **12.21** |
| FairViT | 22.03 | 21.89 | 22.01 |

Table 5: Training time per epoch (in seconds) for FSCL, Vanilla and FairViT on the task **Y:** Attraction, **S:** Gender. FSCL(80), Vanilla and FairViT are trained on the first 80 celebrities, and FSCL(Full) on the entire dataset. One run was conducted for the FSCL model, and three runs were conducted for Vanilla and FairViT (average is shown).

| Model | Y: Attraction, S: Gender |
|---|---|
| | time (sec) |
| Vanilla | 12.20 |
| FairViT | 22.03 |
| FSCL(80) | **10.14** |
| FSCL(Full) | 1080.06 |

## 4.2 Results beyond original paper

### 4.2.1 Additional datasets

Due to the lack of reproducible evidence supporting the authors' claims, we extended our evaluation to the UTKFace dataset to further validate our findings by comparing FairViT with the Vanilla ViT model.

By intentionally creating an imbalanced training dataset, we investigated the models' accuracy and fairness in predicting gender, using race as the sensitive attribute. As the dataset contains five race categories, we simplified the analysis by binarising the sensitive attribute into "white" and "non-white" groups. As demonstrated in Table 6, no improvements were observed in any metric when using FairViT. The differences between FairViT and the Vanilla model were within the margin of statistical error.

### 4.2.2 Evaluation of the proposed loss function on additional architectures

To assess the extensibility and effectiveness of the proposed distance-based loss function in different settings and architectures, we conducted additional experiments using two architectures: a pre-trained ResNet50 (He et al., 2015) and a Graph Neural Network (GNN). For graph classification, we employed a simple GNN

Table 6: Accuracy (ACC), Balanced accuracy (BA), Equal Opportunity (EO) and Demographic Parity (DP) for one classification task on UTKFace, averaged over two independent runs, for the Vanilla and FairViT models. **Y** denotes the target attribute and **S** denotes the sensitive attribute.

| Model | **Y**: Gender, **S**: Race | | | |
|---|---|---|---|---|
| | ACC% ↑ | BA% ↑ | $EO_{e-2}$ ↓ | $DP_{e-1}$ ↓ |
| Vanilla | **94.48 ± 0.07** | **94.47 ± 0.07** | **8.00 ± 0.42** | **0.837 ± 0.084** |
| FairViT | 94.20 ± 0.36 | 94.25 ± 0.36 | 9.17 ± 1.26 | 0.875 ± 0.038 |

consisting of three Graph Convolutional Layers (GCNs). Each model was trained with and without the addition of the distance loss to the standard cross-entropy objective commonly used in classification tasks. Tables 7 and 8 present the obtained accuracy for fine-tuning the pre-trained ResNet model and training the GNN model respectively, both using solely the standard cross-entropy loss and using a combination of the cross-entropy and the proposed distance loss. A supplementary analysis of the effect of varying $\alpha$ and $\gamma$ values for both models is provided in Appendix B.

Using ResNet50, we observed an 1 percentage point improvement in attractiveness prediction and a 3 percentage points improvement in expression prediction when incorporating the distance loss. For graph classification, accuracy improved by 1.17 percentage points on the AIDS dataset and by 2.09 percentage points on the PROTEINS dataset, under the optimal configuration of $\alpha$ and $\gamma$. These results offer partial support for the authors' claim that the distance loss contributes positively to overall model accuracy.

Table 7: Accuracy results for training the ResNet50 model without and with the distance loss. Shown is the average of three independent runs. Highlighted is the best result.

| Model | **Y:** Attraction | **Y:** Expression |
|---|---|---|
| | ACC% ↑ | ACC% ↑ |
| $L_{ce}$ | 76.44 ± 0.46 | 87.41 ± 1.59 |
| $L_{ce} + L_{dist}$ | **77.59 ± 0.34** | **90.59 ± 0.61** |

Table 8: Accuracy results for training a GNN model without and with the distance loss, in the AIDS and PROTEINS datasets. Shown is the average of three independent runs. Highlighted is the best result.

| Model | **AIDS** | **PROTEINS** |
|---|---|---|
| | ACC% ↑ | ACC% ↑ |
| $L_{ce}$ | 82.33 | 67.26 |
| $L_{ce} + L_{dist}$ | **83.50** | **69.35** |

## 5   Discussion

In this study, we investigated the reproducibility of "FairViT: Fair Vision Transformer via Adaptive Masking". Overall, we were unable to reproduce the primary claim made by the authors and only partially replicated one of the remaining two.

The primary claim concerns the ability of adaptive masks to improve accuracy while maintaining fairness for the examined classification tasks. The original paper reported a significantly higher accuracy (5-10 percentage points compared to the Vanilla model) and better fairness metrics for the FairViT model for all three presented classification tasks. However, we observed no clear improvements achieved by FairViT compared to its baselines, neither when adaptive masking was combined with the distance loss nor when it was used independently with standard cross-entropy loss (see Tables 1 and 3). The lack of performance

improvement when using adaptive masks could likely be attributed to the fact that the group-specific masks may not be highly informative, as they are averaged before being applied and only receive small updates during training, thus leading to largely unaffected model performance.

Examining the proposed distance loss function, we were only able to partially validate the second claim. As presented in Table 3, our results do not show a consistent accuracy improvement when using the distance loss for the CelebA dataset in the context of FairViT. However, to further examine the validity of this claim, we studied the effect of the proposed loss function in two additional architectures, namely a ResNet and a GNN. In this case, the distance loss showed a greater benefit, yielding an accuracy improvement of approximately 1-3 percentage points (see Tables 9, 10, 11 and 12). These results suggest that the distance loss can enhance accuracy, though its effectiveness appears architecture-dependent. Further studying its effect on different tasks and models would therefore be of great interest for future work.

Our results do not support the authors' claim that FairViT's computational cost is comparable to that of the Vanilla ViT. In particular, in Table 4 we can see that its time cost is almost double per epoch. Additionally, when comparing the time taken per epoch for the training of FairViT with that of the FSCL model, we did not observe the difference described in the original paper, when training both models on the same data. In contrast, our experiments showed that the per-epoch training time for FSCL was comparable to - or in some cases lower than - that of FairViT. This stands in stark contrast to the original paper, which reported FSCL as being approximately six times slower than FairViT. We hypothesise that this discrepancy may stem from differences in the measurement methodology, dataset size, or model configuration. We also note that the original paper did not specify whether FSCL was pre-trained on the full training set or only on the subset used for FairViT, which may have further contributed to the mismatch in reported performance and training times. Importantly, FSCL is trained from scratch and does not rely on pre-trained weights, which suggests it should be trained on the full dataset rather than fine-tuned on a smaller subset, as was done in the original study, according to subsequent correspondence with the authors.

## 5.1  What was easy

While trying to reproduce the experiments of the authors, we found certain aspects of the reproduction process that were straightforward. The datasets were easily accessible and pre-processing was straightforward. Additionally, the code was publicly available on GitHub, and the training hyperparameters for FairViT were reported in the original paper.

## 5.2  What was difficult

During our reproducibility study, we encountered several challenges and inconsistencies related to the environment, code and documentation provided by the authors. More specifically, some packages specified in the environment setup were either unavailable or incompatible with the provided code. Additionally, the instructions provided in the GitHub repository for integrating the dataset with the code did not correspond to the provided data parsing code sections, while the required dataset structure was not clearly specified in either the paper or the code repository.

We also noticed discrepancies between the default hyperparameter values in the code and the values mentioned in the paper. Certain components, such as the Gradient Attention Rollout implementation, were absent from the codebase and had to be integrated by us. In addition, the proposed adaptive masking mechanism was used in several parts of the architecture that were not documented in the paper, making the understanding of the model's behaviour challenging.

Moreover, the ablation study in Table 3 of the original paper lacked information about the initialisation of the non-trainable masks, while code adaptations were needed in order to support running all ablations without hard-coding changes in the code. Finally, no details about how the Vanilla Vision Transformer was executed were provided, and the dataset size used for training the FSCL model was not specified in the original paper.

### 5.3 Communication with original authors

The correspondence with the authors was adequate. They responded helpfully to questions concerning the CelebA dataset structure and splits, the way the baseline models were run, the mechanism of the adaptive masking and the reported time cost.

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

## A  Grad Attention Rollout

Gradient Attention Rollout (GAR) (Abnar & Zuidema, 2020) is used for the interpretation of the attention mechanism in ViTs, through the generation of heat maps that measure the contribution of each input patch withing to the attention mechanism's final output. It is mathematically defined as:

$$A_l = \begin{cases} A_l(\mathbf{x}) \frac{\partial \hat{y}}{\partial A_l(\mathbf{x})} A_{l-1}, & \text{if } l > 0 \\ A_l(\mathbf{x}) \frac{\partial \hat{y}}{\partial A_l(\mathbf{x})}, & \text{if } l = 0 \end{cases} \tag{11}$$

where $A_l$ represents the GAR at the $l$-th layer of the transformer, and $A_l(\mathbf{x})$ is the attention map at that layer for the input $\mathbf{x}$. The heat map is constructed by assigning the value $A_N^{0,i}$ to the $i$-th patch in the image, where $A_N$ indicates the GAR of the final layer, and $A_N^{0,i}$ corresponds to the element in the first row and $i$-th column of the matrix $A_N$.

## B    The effect of hyperparameters $\alpha$ and $\gamma$ on Distance Loss

Tables 9 and 11 illustrate the impact of varying the hyperparameter $\alpha$, which regulates the contribution of the distance loss to the overall training loss, on the performance of the ResNet50 and GNN models. The results indicate that the ResNet50 model tends to benefit more from higher values of $\alpha$, while the GNN model performs better when the distance loss has a lower relative weight.

Tables 10 and 12 present the achieved accuracy for different values of the hyperparameter $\gamma$, for the model with the best performing $\alpha$, for the ResNet50 and GNN models. A value of $\gamma$ equal to 0.7 leads to the best performance for both tested experiments for the ResNet50 case. For the GNN model, results are dataset-dependent, with a larger value of 0.7 or 0.9 being preferable for the AIDS dataset and a smaller value of 0.1 leading to significantly higher accuracy for PROTEINS.

Table 9: Impact of $\alpha$ in the distance loss for ResNet50. Shown is the average of three independent runs. Highlighted is the best result.

| $\alpha$ | **Y:** Attraction ACC% | **Y:** Expression ACC% |
|---|---|---|
| No distance loss / 0 | $76.44 \pm 0.46$ | $87.41 \pm 1.59$ |
| 0.001 | $76.89 \pm 0.35$ | $88.53 \pm 0.54$ |
| 0.01 | $76.57 \pm 0.70$ | $89.33 \pm 0.16$ |
| 0.1 | $76.72 \pm 0.90$ | $\mathbf{90.13 \pm 1.44}$ |
| 1 | $\mathbf{77.33 \pm 0.71}$ | $90.01 \pm 1.87$ |

Table 10: Impact of $\gamma$ in the distance loss. Shown is the average of three independent runs. Highlighted is the best result.

| $\gamma$ | **Y:** Attraction ACC% | **Y:** Expression ACC% |
|---|---|---|
| No distance loss | $76.44 \pm 0.46$ | $87.41 \pm 1.59$ |
| 0.1 | $75.91 \pm 0.92$ | $89.66 \pm 1.67$ |
| 0.3 | $76.23 \pm 1.53$ | $88.88 \pm 1.01$ |
| 0.5 | $77.33 \pm 0.68$ | $90.01 \pm 1.08$ |
| 0.7 | $\mathbf{77.59 \pm 0.34}$ | $\mathbf{90.59 \pm 0.61}$ |
| 0.9 | $77.47 \pm 0.60$ | $89.66 \pm 1.05$ |

Table 11: Impact of $\alpha$ in the distance loss for the GNN in the AIDS and PROTEINS datasets. Shown is the average of three independent runs. Highlighted is the best result.

| $\alpha$ | **AIDS** ACC% | **PROTEINS** ACC% |
|---|---|---|
| No distance loss / 0 | 82.33 | 67.26 |
| 0.001 | **83.00** | 66.97 |
| 0.01 | 81.17 | **68.75** |
| 0.1 | 82.00 | 66.67 |
| 1 | 80.50 | 66.67 |

Table 12: Impact of $\gamma$ in the distance loss for the GNN in the AIDS and PROTEINS datasets. Shown is the average of three independent runs. Highlighted is the best result.

| $\gamma$ | AIDS ACC% | PROTEINS ACC% |
|---|---|---|
| No distance loss | 82.33 | 67.26 |
| 0.1 | 83.17 | **69.35** |
| 0.3 | 83.33 | 66.37 |
| 0.5 | 83.17 | 64.29 |
| 0.7 | **83.50** | 66.37 |
| 0.9 | **83.50** | 64.58 |

