# OpenReview forum: "Reproducibility Study of "FairViT: Fair Vision Transformer via Adaptive Masking""
_TMLR — Rejected by TMLR_

### Review · Reviewer_pgAU · 2025-03-15

**Summary Of Contributions:**

The paper presents a reproducibility study of the FairViT model proposed by Tian et al. (2024), which introduced adaptive masking and a distance-based loss to improve fairness and accuracy in Vision Transformers (ViTs). The study attempts to replicate key claims of the original FairViT paper using CelebA and UTKFace datasets. The authors also extend their analysis with additional experiments, including:

1. Reproducing the original FairViT results on CelebA.
2. Testing FairViT on a new dataset (UTKFace) to assess generalizability.
3. Ablation study to isolate the effect of adaptive masking.
4. Applying the proposed distance-based loss to ResNet50 and GNN architectures to examine its extensibility.

The key finding of the paper is that adaptive masking does not provide significant improvements in fairness or accuracy, and FairViT's computational cost is higher than the original claim. The distance-based loss shows marginal improvements in accuracy but is not always statistically significant.

**Audience:**

Yes

**Broader Impact Concerns:**

1. Potential Misrepresentation of Fairness Outcomes
- The paper's reproducibility study reports significantly different fairness metrics compared to prior research, including unexpected high EO  and DP values.
- If these results are not properly validated, they could mislead future researchers into drawing incorrect conclusions about the effectiveness of fairness-aware techniques like FSCL and FairViT.
- The authors should ensure statistical robustness and examine dataset demographic balance to avoid mischaracterizing fairness interventions in vision transformers.

2. Transparency in Computational Cost Reporting
- The study's computational cost analysis lacks transparency and does not fairly compare FairViT with FSCL across all scenarios.
- Without full training-time comparisons, researchers may misinterpret the efficiency trade-offs of fairness-aware ViT models.

---

### Recommendation
Given these ethical concerns, the authors should include a Broader Impact Statement discussing:
- The fairness metrics reported in the study deviate significantly from prior research, raising concerns about the reproducibility and validity of fairness-aware techniques like FSCL and FairViT.
- The lack of dataset demographic balance analysis may lead to mischaracterization of fairness interventions, potentially influencing future fairness evaluations.
- The computational cost analysis is incomplete, as it does not provide consistent comparisons between FairViT and FSCL across all relevant scenarios.
- The absence of full training-time comparisons may result in misleading interpretations of the efficiency trade-offs in fairness-aware ViT models.

By addressing these aspects, the paper will better align with responsible AI research practices and fairness-aware ML principles.

**Claims And Evidence:**

Yes

**Requested Changes:**

1. Improve Experimental Reliability and Reproducibility

- Clarify the discrepancy in fairness results: The fairness metrics reported in Table 1 contradict prior studies, including independent replications of FSCL. The authors should carefully re-examine their experimental setup, data preprocessing steps, and metric computation to identify potential inconsistencies.

- Address unexpectedly high fairness metric values: The reported EO exceeding 0.2 and DP surpassing 0.4 in the Y: Attraction, S: Gender setting is significantly higher than expected. The authors should conduct a sanity check on their fairness computation pipeline to ensure no unintended biases or errors.

- Increase the number of independent runs: The authors report results using only three runs, which is insufficient to account for stochastic variations in model training. To improve statistical robustness, they should conduct at least 5–10 independent runs and report confidence intervals.

- Discuss dataset demographic distributions: Given the sensitivity of fairness evaluation to dataset balance, the authors should explicitly report demographic distributions (e.g., gender ratios, race proportions) and confirm whether any imbalance exists. If imbalanced, they should conduct stratified sampling or include additional fairness-aware normalization techniques.

2. Provide a More Comprehensive Computational Cost Comparison

- Include FSCL in all computational cost comparisons: The study presents a partial comparison of FairViT’s efficiency, but does not consistently compare FairViT against FSCL across all relevant scenarios. The authors should perform a full computational efficiency analysis, directly comparing FairViT and FSCL under identical conditions.

- Clarify the original claim regarding computational efficiency: The FairViT paper did not claim superior efficiency over Vanilla but rather comparable efficiency. The authors should avoid misinterpreting the original claim and ensure their comparisons are framed correctly.

- Extend training time evaluation: The current efficiency analysis is based on a 20-second training window, which is insufficient to evaluate model convergence. The authors should report complete training time per epoch to provide a meaningful computational cost comparison.

- Account for model initialization and loading overhead: Since a significant portion of the reported 20-second runtime may be consumed by initialization rather than actual training, the authors should separate model loading time from training time for a more accurate efficiency evaluation.

3. Provide a Justification for the Failure of Adaptive Masking

- Investigate the underlying causes of masking failure: The study empirically finds that adaptive masking does not improve fairness, yet provides no theoretical explanation. The authors should explore whether this failure is due to limitations in the method or suboptimal implementation choices.

4. Improve Robustness and Generalization of the Evaluation

- Expand dataset evaluation: The study only examines the two datasets (CelebA and UTKFace), which limits the generalizability of the findings. The authors should evaluate their claims on a broader range of datasets, such as FairFace, RFW, or real-world medical datasets.

- Test across multiple architectures: The current study only evaluates ViTs, whereas fairness-aware techniques should ideally be robust across multiple architectures. The authors should test adaptive masking on CNNs and GNNs to determine whether the method generalizes beyond transformers.

- Assess cross-domain generalization: The fairness results may be dataset-specific. The authors should evaluate whether adaptive masking remains effective under distribution shifts or demographic imbalances to validate its real-world applicability.

**Strengths And Weaknesses:**

Strengths:

1. Reproducibility Effort: The paper follows the original FairViT methodology, utilizing the same dataset and evaluation metrics. The authors also extend their analysis by incorporating a new dataset, conducting ablation studies, and evaluating the generalizability of the model, providing additional interpretations beyond the original work.
2. Transparency and Open Science: The authors provide publicly available code, which is crucial for reproducibility studies.

Weaknesses:

1. Questionable Experimental Reliability and Reproducibility
- The results in Table 1 indicate that FSCL and FairViT perform worse than Vanilla in fairness metrics, which contradicts not only the original FairViT paper [2] but also a lot of prior studies [1,3,4,5]. Independent replications of FSCL, including those conducted in previous research, have consistently demonstrated its ability to improve fairness over Vanilla. The observed discrepancies cast serious doubt on the reproducibility of the reported fairness results.
- Moreover, the reported fairness metrics (e.g., EO exceeding 0.2 and DP surpassing 0.4 in the Y: Attraction, S: Gender setting) are significantly higher than expected, which seems either not optimized or misleading dataset dividing choice, especially in CelebA. These unexpectedly high values suggest potential issues with either data preprocessing, experimental design, or metric computation that the authors do not sufficiently address.
- The authors provide only three independent runs to compute average results, which is insufficient to account for stochastic variations in model training. Given the well-known instability in fairness-sensitive learning, a more rigorous study should include a higher number of repetitions to ensure statistical robustness.
- There is no discussion on dataset imbalance, which is a critical factor in fairness evaluations. If the dataset used in their experiments contains a disproportionately high number of male or female samples, it could significantly skew fairness results. The authors should explicitly clarify whether demographic distributions were balanced across different sensitive attributes during their experiments.

2. Misleading and Incomplete Computational Cost Comparisons
- While the paper provides a partial comparison of FairViT’s computational cost, it does not consistently compare FairViT against FSCL across all relevant scenarios. Given FSCL’s widespread usage as a fairness-aware model, a comprehensive computational cost analysis—including FairViT and FSCL under identical experimental conditions—is essential.
- The original FairViT paper never claimed that it was significantly superior in efficiency compared to Vanilla, but rather that it was comparable. The authors of this reproducibility study misrepresent the original claim by suggesting that FairViT should have outperformed Vanilla computationally.
- The reported computational cost analysis is flawed due to biased experimental settings. The study only measures training time over a short 20-second window, which is insufficient to reach a fully fine-tuned model.
- Given that model initialization and loading time can take up a significant portion of the reported 20 seconds, the experimental setup does not accurately reflect real-world training time. Instead, the study should report full training time per epoch to provide a fair and meaningful comparison.

3. Lack of Proper Explanation, Theoretical Analysis to Explain the Failure of Adaptive Masking
- The study provides no theoretical justification for this observation. Without deeper analysis, it remains unclear whether the failure of adaptive masking is due to suboptimal implementation choices.
- A proper fairness analysis should include:
  - Feature visualization to examine whether the learned representations effectively reduce bias.
  - Layer-wise activation analysis to check whether certain layers in FairViT suppress or amplify sensitive information compared to Vanilla models.
  - Gradient analysis to verify if the optimization process is impacted by masking.
  - Demographic Distribution, which is the proportion of different population groups (such as gender, race, age, geographic location, etc.) within a dataset.
- Without these deeper analyses, along with the issues within the experiments themselves, the study remains purely empirical and lacks insight into the underlying factors affecting the model's performance.

[1] Park, Sungho, et al. "Fair contrastive learning for facial attribute classification." Proceedings of the IEEE/CVF Conference on Computer Vision and Pattern Recognition. 2022.

[2] Tian, Bowei, Ruijie Du, and Yanning Shen. "FairViT: Fair Vision Transformer via Adaptive Masking." European Conference on Computer Vision. Cham: Springer Nature Switzerland, 2024.

[3] Zhang, Fengda, et al. "Fairness-aware contrastive learning with partially annotated sensitive attributes." The Eleventh International Conference on Learning Representations. 2022.

[4] Park, Sungho, and Hyeran Byun. "Fair-VPT: Fair Visual Prompt Tuning for Image Classification." Proceedings of the IEEE/CVF Conference on Computer Vision and Pattern Recognition. 2024.

[5] Nielsen, Stefan K., and Tan M. Nguyen. "An Attention-based Framework for Fair Contrastive Learning." arXiv preprint arXiv:2411.14765 (2024).

---

> ### Author Response · Authors · 2025-04-02
>
> We would like to thank the reviewer for their valuable and detailed comments. We provide a point-by-point response below and have updated the paper accordingly, including the addition of standard deviations to our reported results to improve clarity and statistical transparency.
>
> Regarding weaknesses and requested changes:
>
> 1.1-1.2: Regarding FSCL, which follows a two-stage training procedure, we first pretrained the ResNet backbone using contrastive learning on the full dataset, in line with the original FSCL code. We then fine-tuned the classification head using only the images of the first 80 individuals in CelebA, as these were used for training in the FairViT paper. This choice was made to align our setup with FairViT’s experimental design. We acknowledge that this differs from the typical FSCL setup and can explain the observed discrepancies in fairness metrics.
>
> 1.3: We used three independent runs per model to match the setup of the original FairViT paper, which also reports results averaged over three seeds. However, we agree that reporting variability is important, and we have now included standard deviations in the revised manuscript. While more runs would certainly enhance statistical robustness, we found the variance to be low enough that it did not essentially affect the conclusions of our study.
>
> 1.4: The dataset split and setup were based on recommendations we received from the authors of FairViT during direct correspondence. We adopted this approach to replicate the original experiments as faithfully as possible.
>
> 2.1-2.2: FSCL requires contrastive pretraining over the entire dataset, which is computationally very expensive. Pretraining FSCL for 100 epochs, as recommended in its original paper, takes approximately 30 hours on an NVIDIA A100 GPU. Since FairViT’s evaluation spans three different experiments, each requiring three runs, reproducing FSCL across all settings would have required over 12 days of GPU time for a single baseline - beyond our resource constraints.
> In our experiments, FairViT did not outperform the Vanilla method in terms of metrics, while requiring nearly double the computational cost. Since FairViT struggled against the Vanilla ViT in both terms of metrics and computational cost, we argue that evaluating FSCL against FairViT would not provide deeper insights into what is going on with FairViT, also given our limited resources.
> Furthermore, we did not claim that FairViT is superior in computational efficiency; rather, we investigated whether the efficiency was comparable, in line with the original paper’s claim. Our results, presented in Table 4, suggest that FairViT is significantly slower than Vanilla.
> Additionally, we note that the original FairViT experiments used a small subset of CelebA (images from the first 80 individuals), but the setup used for FSCL was not specified in the original paper.
>
> 2.3-2.4: We measured training time per epoch by starting the timer immediately before dataloader iteration and stopping it once the epoch completed. This approach excludes model loading and initialisation time, aligning with standard practice and the FairViT implementation. We are unsure what the reviewer is referring to with the “20-second training window”, but we would be happy to clarify or adjust our measurement further if this concern is more precisely specified.
>
> 3.1: Our implementation of FairViT was based directly on the official code provided by the original authors, with only minimal changes. As such, we believe that the failure of adaptive masking in our experiments is unlikely to be due to implementation errors.
>
> 3.2-3.3: For this purpose, we have used gradient attention rollout to visualise which regions of the image the model attended to most. This was intended to provide insight into the features influencing classification.
>
> Please refer to our subsequent comment for a detailed response to the remaining concerns.

---

> > ### Author Response · Authors · 2025-04-02
> >
> > 4.1: The primary objective of our study was to conduct a reproducibility study, and since the original FairViT paper evaluated its model only on a single dataset (CelebA), we considered it a meaningful extension and sufficient to include a second dataset (UTKFace) for assessing generalisability. This approach allows us to evaluate the claims made in the original paper and assess whether the findings hold across different datasets, however, since the method did not seem to work on either dataset, we consider it unnecessary to evaluate on even more datasets. Additionally, the distance loss function was tested across multiple architectures and datasets, providing a more comprehensive evaluation.
> >
> > 4.2: We agree that testing adaptive masking across multiple architectures would be an interesting direction for future work. However, because our study's focus was on reproducibility, the paper only applied adaptive masking to ViTs and since we found that adaptive masking did not work as claimed in the original work, our main goal was to validate this result, and not to extend it to other architectures. Given that adaptive masking did not work as expected in the original study, we had limited motivation to extend this to other architectures at this stage. However, the loss function, which we identified as more promising, was tested on multiple architectures and datasets in order to provide more insight into its reproducibility and transferability.
> >
> > 4.3: Regarding the effectiveness of the adaptive masking across domain shifts or demographic imbalances, we have to point out that adaptive masking was not effective in our experiments. The primary goal of our study was to verify the findings of the original work, and since we found that the adaptive masking does not deliver the claimed fairness improvements, we did not consider it further in terms of cross-domain generalisation. Given that the method failed in our replication study, the question of its performance under distribution shifts or demographic imbalances was not something we considered at this stage. Therefore, we did not focus on cross-domain generalisation since adaptive masking itself is not effective according to our results.
> >
> > We would like to thank the reviewer again for their feedback and would be happy to receive further suggestions if they believe additional clarifications or improvements are needed.
> >
> > Sincerely,
> > The Authors

---

> > > ### Comment · Reviewer_pgAU · 2025-04-14
> > >
> > > Thank you to the authors for their detailed responses and the updates to the manuscript, including the addition of standard deviations and clarifications on FSCL implementation. I appreciate the effort to align experimental setups and the inclusion of UTKFace to test generalizability. However, the deviation from FSCL’s standard setup and limited discussion of dataset imbalance still weaken the fairness conclusions. While I understand the computational constraints, the partial evaluation of FSCL and lack of deeper analysis on adaptive masking limit the study’s interpretability. I encourage the authors to further strengthen the analysis and provide aligned and deeper explorations of the method. Therefore, I believe the paper still requires further exploration.

---

> > > > ### Author Response · Authors · 2025-04-15
> > > >
> > > > Dear Reviewer,
> > > >
> > > > We would like to thank you for your comment. We acknowledge that our approach deviates from the standard FSCL setup, and we agree that a deeper analysis could enhance the interpretability of the study. We would like to clarify that this submission is part of the ML Reproducibility Challenge (MLRC 2025 – https://reproml.org/), which specifically focuses on assessing the reproducibility of published work, by verifying their claims and evaluating the generalisability of their findings. Accordingly, our primary objective was to reproduce the original experiments using the authors' original setup, which explains the deviation from the conventional FSCL framework.
> > > >
> > > > Sincerely,
> > > > The Authors

---

### Review · Reviewer_RE3D · 2025-03-16

**Summary Of Contributions:**

This paper
- Evaluates FairViT [1], a Vision Transformer addressing fairness issues using adaptive masking and distance-based loss.
- Conducts a reproducibility study on CelebA and UTKFace datasets, comparing FairViT with Vanilla ViT and FSCL baselines.
- Finds adaptive masking ineffective for fairness/accuracy, while distance-based loss shows partial effectiveness.
- Highlights FairViT’s higher computational cost compared to baselines.

[1] FairViT: Fair Vision Transformer via Adaptive Masking. ECCV 2024.

**Audience:**

Yes

**Claims And Evidence:**

Yes

**Requested Changes:**

1. I strongly recommend that the authors thoroughly review the paper and correct typos and informal writing styles. This is an academic paper, not a blog. Please treat it with the seriousness it deserves.
2. Please explain why your reproduced results differ significantly from those reported in numerous previous studies.
3. Would it be possible to include more established and widely recognized fairness algorithms as benchmarks for comparison?

**Strengths And Weaknesses:**

Strengths:
- Critical evaluation of FairViT’s fairness and efficiency.
- Validates findings on multiple datasets, enhancing robustness.
- Provides insights into trade-offs between fairness improvements and computational costs.

Weaknesses:
1. Unprofessional writing and poor presentation:
    - incorrect quotation mark "" in the title and Section 3.5. The correct form is `` '' in latex.
    - a strange and unusual format at the end of the introduction chapter ([Reproducibility Study], [Extended Work]).
    - sometimes only the first word of a chapter is capitalized, and sometimes all words are capitalized.
    - G, H in Section 3.1.1 should be $G$ and $H$.
    - $l$ and $h$ do not need to be capitalized in $\operatorname{Attn}_{l,h}$.
    - redundant parentheses in Equation (2).
    - there should be a comma at the end of Equation (7).
    - CO2 in Section 3.6 should be replaced with $CO_2$.
    - FairVit should be FairViT at the beginning of Section 4.
    - the precision of the results is inconsistent, sometimes retaining four decimal places and sometimes retaining five decimal places. For instance, 80.353 in Table 1.
    - there is not a conclusion section.
2. The reason for including the AIDS and PROTEINS datasets is unclear. Given that the paper primarily focuses on computer vision tasks, reproducing graph learning tasks appears unnecessary and unrelated to the main objectives of the study.
3. Wrong citation: In Section 3.5, when introducing the definition of Demographic Parity (DP), the authors cite it as (Hardt et al., 2016), which introduce another fairness definition Equalized Opportunity (EO).
4. Conflict with Existing Work: In many previous studies [1-4], DP and EO on the CelebA dataset are generally less than 0.2. However, the results in this paper show values close to 0.3 or even 0.4, which contradicts the findings of prior work. Based on the experimental results, I question the quality of the reproducibility in this study.

[1] Fair Mixup: Fairness via Interpolation. ICLR 2021.

[2] Enhancing Fairness of Visual Attribute Predictors. ACCV 2022.

[3] Fair Scratch Tickets: Finding Fair Sparse Networks without Weight Training. CVPR 2023.

[4] Utility-Fairness Trade-Offs and How to Find Them. CVPR 2024.

---

> ### Author Response · Authors · 2025-04-02
>
> We thank the reviewer for their thoughtful and detailed feedback. We have carefully revised the manuscript to address the concerns raised, particularly regarding writing quality and formatting. Below we provide responses to each of the key points.
>
> We appreciate the comments regarding presentation, writing style, typographical errors and formatting and have carefully revised the manuscript to address all concerns. Regarding format at the end of the introduction chapter [Reproducibility Study], [Extended Work], this was originally inspired by prior published work in this domain (e.g., [1], [2]). We also acknowledge the absence of a conclusion section. This decision was motivated by the format adopted in several related works in this domain (e.g., [1]–[5]), which similarly omitted a separate conclusion section. Since our study closely follows the structure of prior reproducibility-focused papers, we chose to maintain that format. However, we understand the reviewer’s concern and are open to including a dedicated conclusion section in a future revision if deemed necessary. We would welcome further feedback from the reviewer on whether they believe this addition would meaningfully improve the clarity or presentation of the paper.
>
> Regarding the experiments on ResNet and GNN models, our objective was to explore the potential of the distance-based loss function across different architectures, which we believe could inspire future research. These experiments serve more to highlight the flexibility and applicability of the loss function in different contexts, a direction also proposed by the authors of the original paper as potential future work.
>
> Regarding FSCL, which follows a two-stage training procedure, we first pretrained the ResNet backbone using contrastive learning on the full dataset, in line with the original FSCL code. We then fine-tuned the classification head using only the images of the first 80 individuals in CelebA, as these were used for training in the FairViT paper. This choice was made to align our setup with FairViT’s experimental design. We acknowledge that this differs from the typical FSCL setup and can explain the observed discrepancies in fairness metrics.
>
> We would like to thank the reviewer again for their feedback and would be happy to receive further suggestions if they believe additional clarifications or improvements are needed.
>
> Sincerely,
> The Authors
>
> [1] D. G. Fernández, R.-A. Matișan, A. M. Muñoz, and J. Partyka, ‘Reproducibility Study of “‘ITI-GEN: Inclusive Text-to-Image Generation’”’, Transactions on Machine Learning Research, 2024.
>
> [2] I. Skylitsis, Z. Feng, I. Nasim, and C. Niessink, ‘Reproducibility Study of “‘Robust Fair Clustering: A Novel Fairness Attack and Defense Framework’”’, Transactions on Machine Learning Research, 2024.
>
> [3] A. Vasilcoiu, T. H. F. Stessen, T. Kersten, and B. Helvacioglu, ‘[Re] GNNInterpreter: A probabilistic generative model-level explanation for Graph Neural Networks’, Transactions on Machine Learning Research, 2024.
>
> [4] M. Hamar, M. Krastev, K. D. Hristov, and D. Beglou, ‘[Re] Explaining Temporal Graph Models through an Explorer-Navigator Framework’, Transactions on Machine Learning Research, 2024.
>
> [5] N. Midavaine, G. H. T. Go, D. Canez, I. Simion, and S. Chatterji, ‘[Re] On the Reproducibility of Post-Hoc Concept Bottleneck Models’, Transactions on Machine Learning Research, 2024.

---

> ### Comment · Reviewer_RE3D · 2025-04-08
>
> Thank you for your responses and revising the manuscript. I have updated the "Claims And Evidence" part. However, after reading the review from other reviewers, while I expect some deeper understanding of the fairness literature, I have not been inspired by this paper. Therefore, I also update the "Audience" part. I believe that "Results beyond original paper" can be further extended to make this paper more interesting. The writing can also be further improved in future iterations of this paper.

---

> > ### Author Response · Authors · 2025-04-15
> >
> > Dear Reviewer,
> >
> > We would like to thank you for your comment and updates. We acknowledge that our study does not introduce methodological innovations, which may limit its interest to a broader audience, and we agree that the "Results beyond original paper" section can be further extended. However, we would like to clarify that our submission is part of the ML Reproducibility Challenge (MLRC 2025 – https://reproml.org/), which specifically focuses on assessing the reproducibility of published work, by verifying their claims and evaluating the generalisability of their findings. In line with this objective, our primary focus was on reproducing the original paper.
> >
> > Sincerely,
> > The Authors

---

> > > ### Comment · Reviewer_RE3D · 2025-04-16
> > >
> > > I see. I appreciate the author’s response and the additional details provided, especially the reference to MLRC 2025. Based on this information, I have updated my "Audience" to 'Yes.' However, I still believe that this paper has room for improvement, and I hope you can address them in the future to inspire further advancements in this field.

---

### Review · Reviewer_3pdo · 2025-03-24

**Summary Of Contributions:**

This paper reproduces and extends FairViT which is proposed in the findings of the ECCV 2024. The authors evaluate the FairViT model on CelebA and UTKFace datasets, as well as on additional architectures like ResNet50 and GNNs. The results indicate that the adaptive masking method does not provide the claimed improvements in fairness or accuracy.

**Audience:**

No

**Claims And Evidence:**

Yes

**Requested Changes:**

see weaknesses

**Strengths And Weaknesses:**

**Strengths**
1. The paper replicates the original experiments and conducts extended evaluations across multiple datasets and architectures.
2. The authors employ a range of fairness and performance metrics.
3. The code is publicly released.

**Weaknesses**
1. While the reproducibility effort is appreciated and the implementation is thorough, this submission lacks the qualities of a publishable research paper. It does not present a novel idea, has poor academic writing and structure, and reads more like an internal project report. Unless significant methodological contributions are added and the presentation is substantially improved, this work is better suited for release on platforms like GitHub or arXiv as an open reproducibility project.
2. The manuscript is poorly formatted, with incomplete or misaligned tables, missing table lines, and inconsistent figure labeling.
3. The authors fail to offer deep analysis on why the adaptive masking did not work as claimed. The discussion is superficial, merely attributing the failure to weak gradient signals or averaging effects without exploring architectural or theoretical causes.
4. The experiments on ResNet and GNN models are loosely connected to the original FairViT framework. There is no discussion on why these architectures are chosen or how the insights from FairViT transfer to them.

---

> ### Author Response · Authors · 2025-04-02
>
> We thank the reviewer for their thoughtful and detailed feedback. Below we provide responses to each of the key points.
>
> 1: As a reproducibility study, our primary goal was not to introduce a novel method, since the proposed method did not work as expected, but to test and validate the claims made in the original FairViT paper and extend the proposed methods on different datasets and architectures. While we acknowledge that the paper does not introduce a purely novel idea, we believe the study offers valuable insights, particularly in showing the limitations of the proposed method. By clearly demonstrating that adaptive masking did not work as expected, and by identifying the partial effectiveness and extensibility of the distance-based loss across different architectures, we offer a meaningful addition to the existing literature. Our work does not aim to expand on the method but to critically assess its effectiveness, but future research can build on this foundation to explore new directions or improvements.
>
> 2: We appreciate the reviewer’s feedback regarding the academic writing and formatting. In response, we have rewritten several sections of the paper to improve clarity, structure, writing and presentation quality and a revised version has been submitted.
>
> 3: We appreciate the reviewer’s suggestion for a deeper analysis regarding the failure of adaptive masking. In our current study, we focused primarily on evaluating whether the methods reproduced the claimed improvements in fairness and accuracy. As the adaptive masking mechanism did not yield the expected results across multiple settings, we decided to focus on evaluating the methods of the original paper on additional datasets and assess the potential of the distance loss, which yielded more promising results, across different architectures. We provided, however, a preliminary discussion pointing to possible factors due to which the adaptive masking may fail, based on code/results inspection, our visualisations and training behaviour. Indeed, we consider a deeper analysis of this aspect to be an important direction for future research.
>
> 4: Regarding the experiments with ResNet and GNN architectures, our intention was to investigate whether the distance-based loss function proposed in FairViT could generalise beyond the Vision Transformer setting. This direction was also suggested by the original FairViT authors as a possible area for future research. While we agree that these experiments are not directly tied to the FairViT model architecture, we believe they provide additional context on the transferability of the loss function and may help inspire further exploration and research.
>
> We would like to thank the reviewer again for their feedback and would be happy to receive further suggestions if they believe additional clarifications or improvements are needed.
>
> Sincerely,
> The Authors

---

> > ### Comment · Reviewer_3pdo · 2025-04-14
> >
> > Thank you for the detailed response.
> >
> > After reading the rebuttal and the comments from other reviewers, I still feel that the contribution of the paper is too limited to warrant acceptance. I understand that the authors did not intend to propose a novel algorithm but test the baseline methods, and I agree that such a goal is valid. However, this paper, in its current form, offers little beyond a straightforward reproduction, which makes it difficult to generate interest or insight for the reader.
> >
> > Given the amount of effort clearly put into the reproduction, I believe the authors must have encountered a range of issues during the process. If the paper could focus on even one of those issues and propose a thoughtful resolution, no matter how small, it would significantly increase the value of this paper and make it much more compelling.

---

> > > ### Author Response · Authors · 2025-04-15
> > >
> > > Dear Reviewer,
> > >
> > > We would like to thank you for your comment. We acknowledge that the contribution of our study is limited, making it difficult to generate interest for a broader audience. Indeed, we encountered several issues during the reproducibility process, and focused on employing an additional dataset and examining whether the distance loss can be beneficial for different architectures. Moreover, we would like to clarify that this submission is part of the ML Reproducibility Challenge (MLRC 2025 – https://reproml.org/), which specifically focuses on assessing the reproducibility of published work, by verifying their claims and evaluating the generalisability of their findings. In line with this objective, our main focus was to verify the claims of the original paper and assess the generalisability of its findings and methods.
> > >
> > > Sincerely,
> > > The Authors

---

### Decision · Action_Editor_Dsmy · 2025-05-04

**Recommendation:** Reject

**Comment:**

This paper presents a reproducibility study of the FairViT model from Tian et al. (2024). The authors perform this reproduciblity study on the CelebA dataset, and also introduce a new dataset, UTKFace, to evaluate generalisability. The authors also extend the approach to the ResNet50 and GNN model architectures.

Reviewers had a number of common concerns that were not adequately addressed during the rebuttal phase: Reviewers felt that the paper is not of general interest to the TMLR audience as it does not provide novel, generalisable insights or analysis. Although reviewers understood that as a reproducibility study, the paper is not required to propose a novel algorithm, new insights from reproducing the method on existing datasets would have been welcome and expected. Moreover, Reviewer pgAU, raised valid concerns about experimental results themselves, pointing out that the authors did not follow the standard FSCL setup when comparing on UTKFace.

Ultimately, all reviewers recommended rejecting the paper, and the action editor agrees with their assessment.

**Audience:**

No. Reviewers felt that the paper is not of general interest to the TMLR audience as it does not provide novel, generalisable insights or analysis. Although reviewers understood that as a reproducibility study, the paper is not required to propose a novel algorithm, new insights from reproducing the method on existing datasets would have been welcome and expected. As a result, a reviewer pointed out that the paper "reads more like an internal project report", and would not be of interest to some part of TMLR's audience.

**Claims And Evidence:**

The claims made in this submission are partially supported.

This paper is a reproducibility study of the FairViT model from Tian et al. (2024). Reviewer pgAU, raised valid concerns about experimental results, pointing out that the authors did not follow the standard FSCL setup when comparing on UTKFace. As a reproduciblity study, it is important to match these details carefully.

**Resubmission Of Major Revision:**

The authors may consider submitting a major revision at a later time.